# Capture ELISA for KPC Detection in Gram-Negative Bacilli: Development and Standardisation

**DOI:** 10.3390/microorganisms11041052

**Published:** 2023-04-18

**Authors:** André Valencio, Miriam Aparecida da Silva, Fernanda Fernandes Santos, Juliana Moutinho Polatto, Marcelo Marcondes Ferreira Machado, Roxane Maria Fontes Piazza, Ana Cristina Gales

**Affiliations:** 1Division of Infectious Diseases, Department of Internal Medicine, Escola Paulista de Medicina, Universidade Federal de São Paulo, São Paulo 04039-032, Brazil; andrebiomedicovalencio@gmail.com (A.V.); marcelo.marcondes@unifesp.br (M.M.F.M.); 2Laboratório de Bacteriologia, Instituto Butantan, São Paulo 05503-900, Brazil

**Keywords:** carbapenemases, *Klebsiella pneumoniae*, antibodies, immunodiagnostic

## Abstract

The detection of KPC-type carbapenemases is necessary for guiding appropriate antibiotic therapy and the implementation of antimicrobial stewardship and infection control measures. Currently, few tests are capable of differentiating carbapenemase types, restricting the lab reports to their presence or not. The aim of this work was to raise antibodies and develop an ELISA test to detect KPC-2 and its D179 mutants. The ELISA-KPC test was designed using rabbit and mouse polyclonal antibodies. Four different protocols were tested to select the bacterial inoculum with the highest sensitivity and specificity rates. The standardisation procedure was performed using 109 previously characterised clinical isolates, showing 100% of sensitivity and 89% of specificity. The ELISA-KPC detected all isolates producing carbapenemases, including KPC variants displaying the ESBL phenotype such as KPC-33 and -66.

## 1. Introduction

The World Health Organization (WHO) has declared antimicrobial resistance (AMR) as one of the top 10 public health threats [1]. Infections caused by antimicrobial-resistant pathogens occur in all geographic regions, increasing the length and cost of hospitalisation, and especially, impacting the morbidity and mortality rates [2,3]. Tacconelli et al. (2018), in collaboration with the WHO, published a list of the bacterial pathogens to be considered as priorities for research and development of new antimicrobials. *Acinetobacter baumannii*, *Pseudomonas aeruginosa*, and Enterobacterales resistant to carbapenems were the top three priority pathogens [4].

Among Enterobacterales, the production of carbapenemases is the main mechanism of carbapenem resistance. Carbapenemases present a great concern worldwide due to their wide variety and dissemination among different species through horizontal gene transfer (HGT). These enzymes belong to classes A, B, and D of the Ambler molecular classification and usually hydrolyse almost all β-lactams [5,6]. Although the frequency of New Delhi Metallo-beta-lactamase (NDM—class B) has increased during the COVID-19 pandemic, *Klebsiella pneumoniae* carbapenemase (KPC—class A) is still the most frequent carbapenemase in many geographic regions [7,8]. The rapid and successful dissemination of *bla*_KPC-_like has been associated with the spread of endemic *K. pneumoniae* CC258 clones. In addition, *bla*_KPC-_like has been acquired by distinct Inc plasmids facilitating its spread [9].

Early adequate antimicrobial therapy has been associated with lower mortality rates. Kumar et al. (2006) reported that for every hour of delay in the adequate prescription of antimicrobials, there was a 7.6% increase in mortality rates of patients with sepsis [10]. Unfortunately, the infections caused by KPC-2-producing *K. pneumoniae* isolates have been associated with high mortality rates [11,12]. New combinations of β-lactamase inhibitor–β-lactams have been developed and are already available for hospital use. However, these combinations only have activity against Gram-negative bacilli (GNB) producers of class A and OXA-48 carbapenemases (class D) [13]. The detection of β-lactamase-encoding genes via the PCR technique, followed by DNA sequencing or whole-genome sequencing, is considered the gold standard for the identification of β-lactamase-encoding genes. Different methodologies were proposed to investigate the production of carbapenemase using a clinical isolate in a short period of time and with a lower cost than PCR. However, tests such as CarbaNP or BlueCarba generally do not differentiate the carbapenemase classes [14,15]. In contrast, immunochromatographic tests have shown high sensitivity and specificity for distinguishing the class of most frequent carbapenemases with a rapid turnaround time for results without requiring technical expertise [16]. Unfortunately, these tests are expensive for low- and middle-income countries such as Brazil (about USD 10.00 per test). In addition, some KPC variants have not been accurately detected via immunochromatographic tests [17]. Although KPC variants not identified with such tests usually show resistance to ceftazidime–avibactam, they are susceptible to carbapenems. However, clinical failure has been observed when carbapenems were prescribed as monotherapy for the infections caused by such variants due to the occurrence of a heterogeneous population [18]. To date, no ELISA tests have been developed for the detection of carbapenemases in GNB. Herein, we aimed to develop an immunoenzymatic test to detect KPC production in GNB clinical isolates, given the extensive need for the detection of carbapenemase production in different Enterobacterales.

## 2. Materials and Methods

### 2.1. Expression and Purification of Recombinant KPC (rKPC-2)

For the development of anti-KPC antibodies, the animals (New Zealand rabbit and Balb/c mice) were immunised with purified recombinant KPC-2 enzyme (rKPC-2). The *K. pneumoniae* strain ATCC BAA-1705 (American Type Culture Collection, Manassas, VA, USA) producer of KPC-2 was commercially obtained and subcultured twice on blood agar (Oxoid, Basingstoke, England). The *bla*_KPC-2_ amplicon obtained from *K. pneumoniae* BAA-1705 was cloned in the plasmid pEt26a+. Primers were designed using the restriction sites for Ndel (forward 5 GGTGGTCATATGTCACTGTATCGCCGTCTAGTT 3′) and Sall (reverse ACCACCGTCGACCTGCCCGTTGACGCCCAATCCCTCGA 3′). The expression of the rKPC-2 protein was analysed using the *Escherichia coli* strain BL21 (D3E) with the addition of 0.2 mM IPTG (isopropyl-β-D-thiogalactoside) as an expression inducer for 18 h at 20 °C. The osmotic shock was performed to release rKPC-2 from the periplasmic space [19]. Purification was assessed using hydrophobic interaction chromatography with a 5 mL Resource PHE column (GE, Healthcare, Orsay, France) at a flow rate of 5 mL/min. The elution of the ligands was analysed using the linear gradient method with 0% to 100% of the elution solution (50 mM NaH_2_PO_4_) in 20 column volumes (CVs). The molar extinction coefficient of the rKPC-2 was calculated by applying the ProtParam tool and using λ 280 nm wavelength.

### 2.2. Production of Antibodies

The experiments were conducted in agreement with the Ethical Principles in Animal Research, adopted by the Brazilian College of Animal Experimentation, and they were approved by the Ethical Committee for Animal Research of the Butantan Institute (CEUA no 9599160922).

A New Zealand rabbit (60–65 days old) and two Balb/c mice (4 to 6 weeks old weighing 18–22 g) were immunised with rKPC-2 for the development of primary and secondary antibodies. The rabbit was immunised with 100 μg of rKPC-2 adsorbed to aluminium hydroxide adjuvant (10 times the amount of antigen) diluted in PBS, via intramuscular followed by two boosters at 15-day intervals. Mice were immunised subcutaneously, and the immunisation protocols consisted of one immunisation with rKPC-2 (10 µg) adsorbed to aluminium hydroxide adjuvant (10 times the amount of antigen) diluted in PBS followed, four weeks later, followed by two booster injections (rKPC-2, 10 µg) with a 15-day interval.

Once a successful immunisation was observed, the animals’ blood was collected, the mice were euthanised, and the serum purified. Polyclonal antibodies (pAbs) from the rabbit and mice were filtered (0.45 µm) and purified using Protein A and Protein G Sepharose columns (GE, Healthcare, Orsay, France), respectively, both coupled to a liquid chromatography system—ÄKTA Purifier (GE Healthcare, Orsay, France). After dialysis against 20 mM sodium phosphate, they were concentrated using PEG 6000. The presence of Fab and Fc was determined via 15% polyacrylamide gel electrophoresis containing sodium dodecyl sulphate (SDS–PAGE) stained with Coomassie blue R-250. The final immunoglobulin dosage was performed at λ280 nm using a Nanodrop Lite Spectrophotometer (Thermo Scientific, Waltham, MA, USA).

### 2.3. Checkboard Titration

The cross-reactivity and the best concentration between mouse rKPC-2-pAb and rabbit rKPC-2-pAb for detecting specific KPC-2 in the bacterial culture supernatant were determined with a capture ELISA immunoassay using different pAb concentrations ranging from 0.781 to 50 μg/mL, with a predetermined rKPC concentration (10 μg/mL) as antigen.

### 2.4. Capture ELISA Immunoassay

The 96-well MaxiSorp microplates (Nunc^®^, Rochester, NY, USA) were incubated with 6.25 µg/mL of mouse rKPC-2-pAb in a carbonate–bicarbonate buffer, pH 9.6, at 37 °C for 2 h and then further at 4 °C for 16 h. Phosphate-buffered saline (PBS) containing 1% of bovine serum albumin (BSA) (PBS-BSA) was added as a blocking agent and incubated for 1 h at 37 °C. Thereafter, the supernatant of bacterial cultures, PBS, and 10 μg/mL rKPC-2 (negative and positive control, respectively) were incubated for 1 h at 37 °C. Next, 6.25 µg/mL of rabbit rKPC-2-pAb was added and incubated for an additional hour at 37 °C. KPC-2 from samples was then detected with goat anti-rabbit IgG peroxidase (Sigma-Aldrich, St. Louis, MO, USA) diluted 1:5000 in a 0.1% PBS-BSA solution. Reactions were developed with 0.5 mg/mL O-phenylenediamine (OPD; Sigma-Aldrich Co, St. Louis, MO, USA) plus 0.5 μL/mL hydrogen peroxide in 0.05 M citrate–phosphate buffer, pH 5.0, in the dark at room temperature. The reactions were interrupted after 15 min via the addition of 50 μL of 1 M HCl. The absorbance was measured at 492 nm using a Multiskan EX ELISA reader (Labsystems, Milford, MA, USA). At each step, the volume added was 100 μL/well, except in the washing and blocking steps, when the volume used was 200 μL/well. Between incubation periods, the plates were washed three times with PBS-Tween 0.05%. All experiments were carried out in technical duplicates, and the results correspond to three independent experiments (biological replicates).

### 2.5. Bacterial Sample Preparation

Thirteen isolates were selected to screen for the best condition for the isolation of bacterial colonies to be tested using an indirect ELISA (Table 1).

Initially, the isolates were subcultured using Mueller–Hinton agar supplemented with ceftazidime 2 μg/mL and incubated at 37 °C for 18 h and then submitted to four different protocols as follows: (i) direct colonies (DCs): A 10 µL loop of bacterial colonies was suspended in 500 µL of 1% PBS (140 mM NaCl; 9 µM Na_2_HPO_4_; 2 µM NaH_2_PO_4_ + H_2_O; pH 7.4) in a 1.5 mL microtube; (ii) pre-inoculum (PD-OD1): Two or three colonies were inoculated in 6 mL of Mueller–Hinton broth (MHB) and incubated at 37 °C for 18 h. After the incubation period, 1% of the bacterial suspension was transferred to a new tube and incubated at 37 °C until reaching an OD of 1.0. Then, 500 µL was transferred to a 1.5 mL microtube and centrifuged at 4100× *g* for 10 min. The supernatant was discarded, and the pellet was resuspended in 500 µL 1% PBS; (iii) pre-inoculum followed by 18 h growth (PD-G): A 1 µL loop of bacterial culture was suspended in 6 mL of MHB and incubated at 37 °C for 18 h. After the incubation period, 1% of the bacterial suspension was transferred to a new tube and incubated at 37 °C for 18 h. Then, 500 µL was transferred to a 1.5 mL microtube and centrifuged at 4100× *g* for 10 min, the supernatant was discarded, and the pellet was resuspended in 500 µL 1% PBS; and (iv) growth in Mueller–Hinton broth (GMHB): A 1 µL loop of bacterial culture was suspended in 6 mL of MHB and incubated at 37 °C for 18 h. Then, 500 µL was transferred to a 1.5 mL microtube and centrifuged at 4100× *g* for 10 min, the supernatant was discarded, and the pellet was resuspended in 500 µL 1% PBS. All tubes were subjected to disruption for 3 cycles of 40 Hz in an ice bath, centrifuged at 16,200× *g* for 10 min, and employed in the ELISA test (Figure 1). The indirect ELISA was performed in duplicate, and the reading was carried out at 492 µm using an ELISA reader (Multiskan—Thermo Sci, Waltham, MA, USA).

### 2.6. Validation of Capture ELISA

The final evaluation of the ELISA-KPC test was carried out using the protocol that presented the highest sensitivity, specificity, and shortest response time for performing the test. Therefore, the performance of the ELISA test was evaluated by testing 109 different isolates previously characterised via PCR and sequencing sanger as follows: 50 isolates carrying different variants of *bla*_KPC_; 3 co-producer carbapenemases (*bla*_KPC-2_/*bla*_BKC-1_); 24 isolates other carbapenemases (non-*bla*_KPC-_like); 19 isolates resistant to carbapenems but non-carbapenemase; 3 isolates producing ESBL; and 10 isolates susceptible to the antimicrobial. Isolates from different species were included in this study (*K. pneumoniae*, *K. oxytoca*, *Enterobacter* spp., *Escherichia coli*, *Proteus mirabilis*, *Shigella flexneri*, *Salmonella enterica* subsp. *Enterica*, *Pseudomonas* spp., and *Acinetobacter baumannii*) (Table 2).

### 2.7. Data Analysis

Data were processed using RStudio to calculate the optimal cut-off, sensitivity, and specificity for each protocol. Student’s *t*-test was used to assess the statistical differences between positive (KPC producers) and negative (non-KPC producers) cases for each method.

## 3. Results

### 3.1. Recombinant bla_KPC-2_

Recombinant *bla*_KPC-2_ was cloned into pEt-26a+. The sequencing result confirmed and demonstrated 100% similarity with the *bla*_KPC-2_ deposited in the NCBI database (Accession number: CP039975.1). The production of rKPC-2 was visualised on an SDS–PAGE gel, where a component of electrophoretic mobility of 32 kDa (Figure 2) was visualised, and then nitrocefin hydrolysis was confirmed. The total yield of purified rKPC-2 protein was 14 mL at the concentration of 705 µg/mL.

### 3.2. Immunisation with rKPC-2 Induced Specific Serum Antibody Response in Rabbit and Mice and No Cross-Reactivity

The generated sera showed high titres after immunisation (at 1/100 dilution (OD 492 µm) >1.0). The titration of rabbit and mouse antibodies was performed using ELISA presenting 1/819,200 (81.92 × 10^4^) and 1/6400 (6.4 × 10^3^) for the rabbit and mice (Figure 3), respectively.

The purification of antibodies using the affinity column recovered 44 mL with a concentration of 65.7 mg/mL and 300µL with a concentration of 1.29 mg/mL for the rabbit and mice, respectively (Figure 4).

The cross-reactivity between pAbs of mouse rKPC-2-pAb and rabbit rKPC-2 for capture ELISA was investigated, and the best concentration chosen was 6.25 μg/mL for both pAbs, which presented the lowest acceptable recognition among the pAbs (cross-reacting) without losing the ability to recognise the studied antigen (rKPC) (Table 3).

### 3.3. Different Bacterial Preparation Induced Different Results in ELISA

The protocol for bacterial sample preparation influenced the ELISA results with very different optimal cut-offs obtained for the four conditions. For the DC protocol, the ROC curve demonstrated an ideal cut-off point of 0.269, with a sensitivity of 75% and a specificity of 77.8%. The pre-inoculum protocols presented cut-offs of 0.486 and 0.437 for PD-OD1 and PD-G, respectively. Both PD-OD1 and PD-G had 100% specificity and 100% sensitivity. The GMHB protocol had a cut-off of 0.159, with 100% sensitivity and 88.9% specificity. Student’s *t*-test showed a statistical difference for PD-OD1, PD-G, and GMHB protocols; however, DC protocol showed *p* < 0.05, not discriminating the producers and non-producers of KPC. The turnaround time for obtaining the ELISA test results was 3–4 h, 22–24 h, 28–30 h, and 40–42 h for DC, GMHB, PD-G, and PD-OD1 protocols, respectively (Figure 5).

### 3.4. Validation of KPC Detection

The GMHB was selected as the best condition for evaluating KPC production via ELISA. A total of 109 bacterial isolates, comprising 53 positives for *bla*_KPC_ and 56 negatives for the *bla*_KPC_, were tested. The variants of KPC (KPC-33 and KPC-66) showing resistance to ceftazidime–avibactam but susceptibility to carbapenems and the KPC-144 variant resistant to ceftazidime–avibactam and carbapenems were evaluated. The KPC-ELISA-GMHB test detected all 53 KPC-producing isolates (50 *K. pneumoniae*, 1 *K. oxytoca*, and 2 *P. aeruginosa*), showing 100% sensitivity and 89.9% specificity with *p* < 0.05 (Figure 6 and Figure 7). Of the 56 non-KPC-producing isolates, 6 (5.5%) were misclassified as KPC producers. False-negative results were observed with *K. pneumoniae* producers of BKC-1 (two isolates) and GES-5 (one isolate); NDM-1-producing *Enterobacter* spp. (one isolate); SPM-1-producing *P. aeruginosa* (one isolate); and a wild-type *K. pneumoniae*.

## 4. Discussion

ELISA is considered the gold standard for diagnosing many infectious diseases [20]. During the last decade, different ELISA tests showing high sensitivity and specificity were developed for the detection of different pathogens such as canine parvovirus (100% sensitivity and 88.4% specificity) and *Schistosoma spindale* (86.7% sensitivity and 90.9% specificity) [21,22]. Chen and colleagues also developed an indirect ELISA test for detecting *K. pneumoniae* in animals; these authors reported that the ELISA test showed higher sensitivity (6.7% increase in positive detection rate) when compared to the agglutination test for the detection of *K. pneumoniae* [23]. Despite the wide use of ELISA tests, we were not able to find any studies conducted thus far that report the development of a similar test for β-lactamase detection. This study reports the first standardisation of an ELISA test for the detection of KPC in GNB isolates. KPC is the most frequent carbapenemase produced by GNB in many geographic regions such as Brazil.

During the standardisation process, we used mouse and rabbit polyclonal antibodies (pAbs), which showed the ability to differentiate KPC-producing isolates from non-KPC producers. pAbs can recognise different epitopes and are produced by different plasma cell clones (differentiated B cells) in response to a specific antigen. The pAbs are an important reagent that can considerably affect the test result. Bradbury (2015) reports that only 0.5% to 5% of pAbs can bind to the intended antigen. In addition, new immunisation is necessary when the amount of antibodies decreases. The production of pAbs may differ from batch to batch, making test reproducibility more difficult [24,25]. Despite these limitations, Ascoli and collaborators reported that pAbs are attractive options due to their clonal diversity and biophysical characteristics such as greater stability. In addition, the possibility of generating pAbs in different animals without the need for specialised equipment and rooms is also very attractive [26]. In agreement with other studies, we also used pAbs for this first standardisation [22,24]. In our experience, the use of pAbs proved to be a good alternative, being able to detect 100% of the KPC-producing isolates and their variants. Easy handling and having the shortest time for pAb generation were decisive in choosing pAbs as reagents. However, the lack of specificity was observed with a higher number of false positives among the isolates producing different β-lactamases, and even among non-β-lactamase producers. The development and standardisation of the KPC-ELISA-GMHB with monoclonal antibodies may be a more appropriate alternative to increase the specificity without losing the sensitivity of the KPC-ELISA-GMHB test.

The cross-reactions observed in the KPC-ELISA-GMHB test may impact clinical decisions on the management of infected patients. The false-positive reactions observed for GES-5- or BKC-1-producing isolates would not significantly lead to the change in antimicrobial prescription, as both enzymes belong to class A carbapenemases. In this case, the new beta-lactamase inhibitor–beta-lactam combinations (IBL-BL), such as ceftazidime–avibactam, are first-line therapeutic agents. In contrast, the false detection of NDM-1- and SPM-1-producing isolates as KPC producers are worrying, as this result could impact the selection of the most appropriate antimicrobial therapy, possibly leading to therapeutic failure [27]. The false detection of non-carbapenemase-producing bacteria is also concerning because it would not allow the de-escalation of antimicrobials, promoting the inappropriate use of antimicrobials and, further, the possible selection of resistant bacterial isolates. To overcome this problem, rapid tests such as CarbaNP or BlueCarba could be initially tested for excluding false-positive isolates.

Although the ELISA test had high sensitivity (100%) and specificity (84%), the incubation period for preparing the bacterial isolate before ELISA testing was an important test limitation. The use of bacterial colonies directly from the growth plate did not show the ability to differentiate KPC from other carbapenemases. Isolate preparation using an initial pre-inoculum showed better sensitivity and specificity; however, the incubation period (28–42 h) was too long. The presence of a single inoculum with incubation for 18 h showed a high sensitivity (100%) but reduced specificity (88.9%) when compared to the pre-inoculum methods (100% for both tests with the pre-inoculum). The reduction in the incubation period (22–24 h) proved to be a better alternative for the clinical use of KPC-ELISA-GMHB. The influence of bacterial sample preparation on the ELISA results could have different explanations. Bacterial growth in different culture media (broth and solid) was evaluated by Fujikawa and Morozumi in 2005. In this study, the authors noted a difference in the number of colony-forming units (CFUs), with a higher number of CFUs in agar than in broth media [28]. Studies carried out by Fortuin and collaborators demonstrated that proteomic differences might occur between the colonies obtained from agar media and those obtained from broth media. The expression of some proteins was higher in solid media, while others were more expressed in broth media [29]. Although bacterial growth was greater on agar media, Warrem and collaborators reported that bacterial colony growth occurs by pushing bacterial cells upward. This process causes the older bacterial cells to be at the top and the younger ones, which were undergoing cell division, at the bottom of the colony [30]. Older bacteria can enter a stationary phase and grow slowly, in addition to important structural changes in the bacterial cell [31]. Jaishankar and collaborators reported that in the stationary phase, there was an increase in the thickness of the bacterial membrane, a decrease in cell space, morphological change (changing from rod to spherical form), and a decrease in protein synthesis [32]. In addition to these characteristics, Kragh and collaborators also noted the presence of bacterial aggregates on agar media, which could not be removed via disruption. These same aggregates could promote tolerance to antimicrobials independent of β-lactamase production [33]. These facts could justify the poor performance of the KPC-ELISA-GMHB test when the bacterial colonies were directly obtained from the growth plate. In broth media, oxygen dispersion is also another important factor influencing bacterial growth. Changes in bacterial metabolism secondary to low oxygen dispersion in broth media have been reported [34]. In our study, the best results for the ELISA test were obtained when a pre-inoculum suspension was used for sample preparation. The inclusion of a second inoculum could have reduced the formation of aggregates and antimicrobial tolerance [33]. Studies evaluating the impact of different culture media (agar and broth) on the expression of β-lactamases are necessary for the better clarification of our results and optimisation of the sample preparation protocol.

The new beta-lactamase inhibitor–beta-lactam combinations such as ceftazidime–avibactam (CAZ-AVI) are recommended as first-line therapy for the treatment of KPC infections [35]. Resistance to CAZ-AVI, especially during or after therapy, has been increasingly reported due to the emergence of KPC-2 variants possessing mutations in the Ω loop such as KPC-33, KPC-66, and KPC-144 [36]. However, these D179Y mutants (KPC-33 and KPC-66) show a reversion of the carbapenem phenotype of resistance becoming susceptible to carbapenems (CAZ-AVI^R^/Carbapenem^s^ phenotype) [37]. Initial studies have reported the clinical success of carbapenems in treating CAZ-AVI-resistant infections caused by such mutants [38,39]. However, sublethal meropenem concentrations may select meropenem-resistant variants in vitro [40]. The presence of a heterogeneous population of wild-type and mutated *bla*_KPC-2_ and the reversibility of the genotypes represent a significant challenge for managing KPC-producing isolates because phenotypic tests such as immunochromatographic, CarbaNP, and BlueCarba fail to recognise the mixed population and often report these isolates as non-carbapenemase-producing isolates, leading to the inadequate prescription of antimicrobial therapy [17,41,42,43]. In our study, the KPC-ELISA-GMHB test detected KPC-2 variants (KPC-33 and KPC-66,), which show CAZ-AVI^R^/Carbapenem^S^ phenotype. It could represent an advantage of the KPC-ELISA-GMHB test over other phenotypic tests for the detection of KPC-2 with a mutation in the Ω loop. The use of polyclonal antibodies in the ELISA platform reduces the cost of the test and increases the KPC detection capacity. The evaluation of polyclonal antibodies in immunochromatographic platforms can also be a promising alternative to the detection of KPC.

## 5. Conclusions

The diagnosis and treatment of bacterial infections have become challenging due to the emergence and spread of multidrug-resistant pathogens harbouring multiple mechanisms of resistance. The presence of gene variants with a different spectrum of activity has become even more challenging. Fast, simple, and accurate tests are of fundamental importance to help antimicrobial stewardship activities in guiding antimicrobial therapy decisions. We proposed the first standardisation protocol for KPC detection using pAbs on an ELISA platform. The ELISA-KPC test showed high sensitivity and specificity, in addition to being the first test besides PCR to detect different KPC variants that have an ESBL profile. The test can be applied to automated systems, reducing human interference and the subjectivity of conventional tests.

## Figures and Tables

**Figure 1 microorganisms-11-01052-f001:**
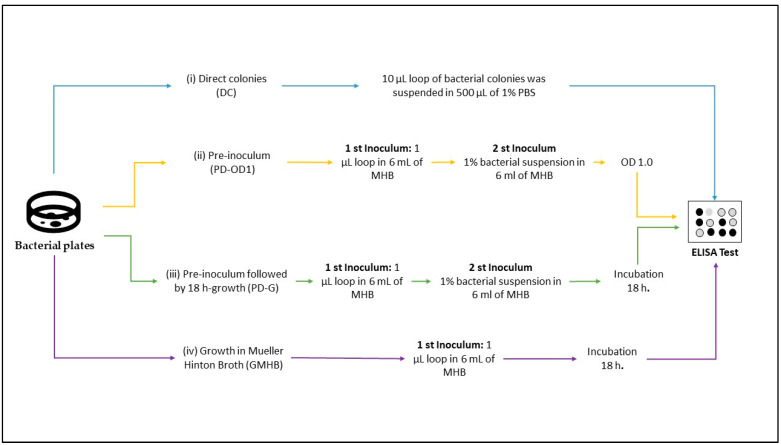
Graphical summary of protocols for the preparation of bacterial isolates before the ELISA test.

**Figure 2 microorganisms-11-01052-f002:**
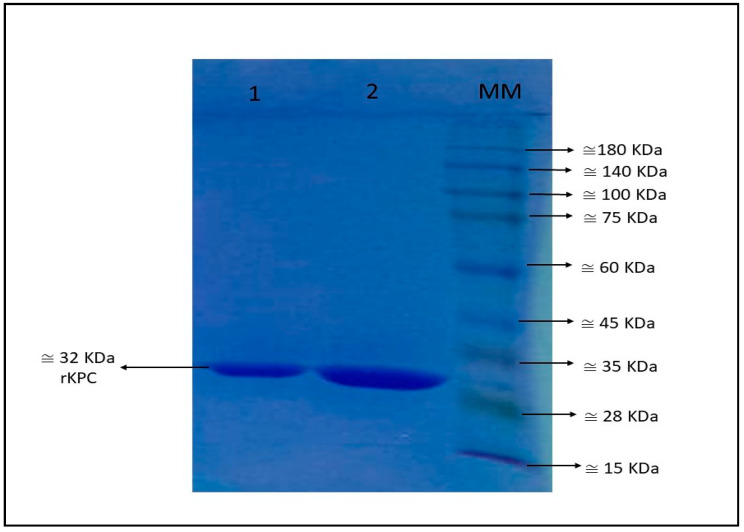
Presence of purified rKPC-2 on SDS–PAGE gel. The top two fractions were collected from AKTA equipment. Legend, 1: Tube 17, 2: Tube 18, MM: molecular marker.

**Figure 3 microorganisms-11-01052-f003:**
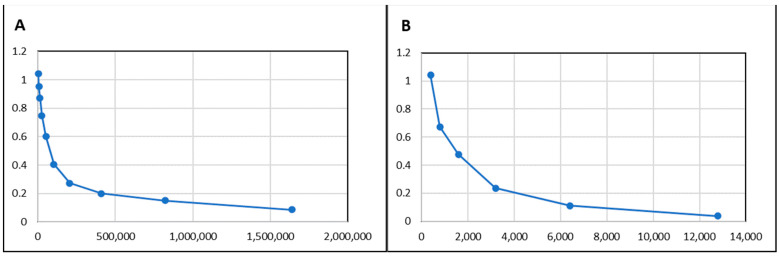
Titration of animal serum containing antibodies against rKPC-2. Legend: (**A**) titre of anti-rKPC-2 antibodies in rabbit serum; (**B**) titre of anti-rKPC-2 antibodies in mouse serum.

**Figure 4 microorganisms-11-01052-f004:**
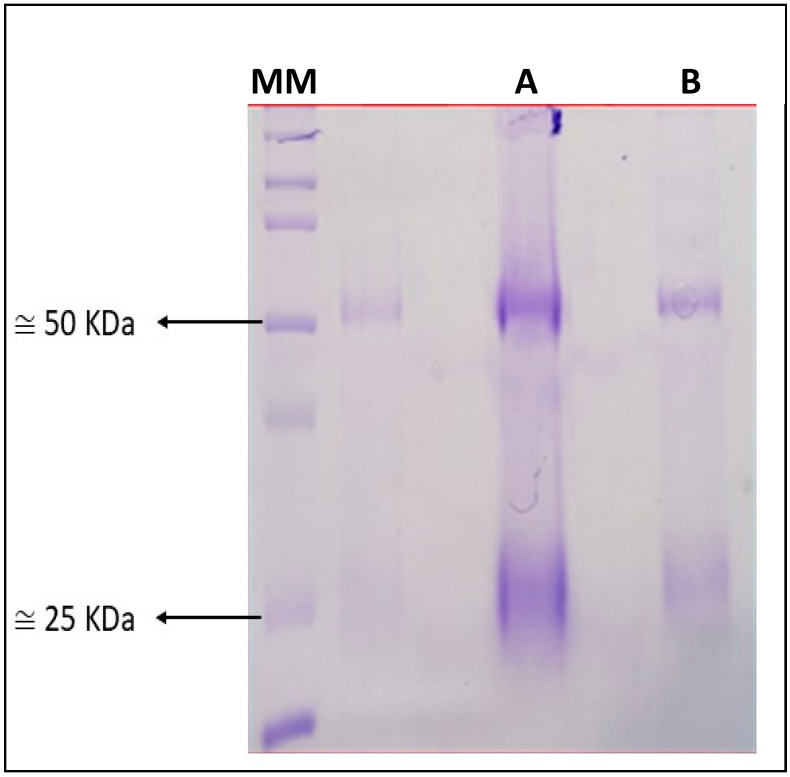
Purification of polyclonal antibodies from (**MM**) molecular marker; (**A**) rabbit and (**B**) mice on SDS–PAGE gel. Antibodies have two subunits, one with 50 KDa and the other with 25 KDa.

**Figure 5 microorganisms-11-01052-f005:**
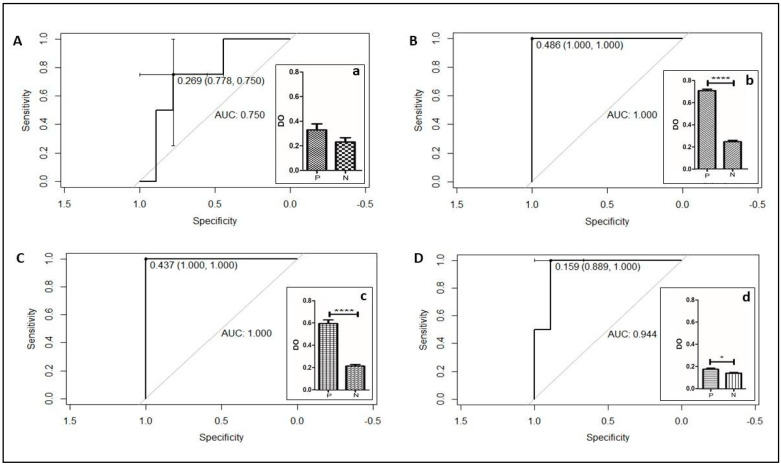
Statistical parameters of the four protocols of bacterial inoculum preparation: (**A**) ROC DC curve with cut-off 0.269, sensitivity: 77.8%, and specificity: 75%; (**a**) *t*-test *p*-value < 0.05; (**B**) ROC PD-OD1 curve, cut-off of 0.486, sensitivity of 100%, and specificity of 100%; (**b**) *t*-test **** *p*-value < 0.0001. (**C**) ROC PD-G curve, cut-off of 0.437; 100% sensitivity, and 100% specificity; (**c**) *t*-test with **** *p*-value < 0.0001; (**D**) ROC GMHB curve, cut-off 0.159, sensitivity 88.9%, and specificity 100%, (**d**) *t*-test * *p*-value < 0.05; legend: P: positive isolates; N: negative isolates.

**Figure 6 microorganisms-11-01052-f006:**
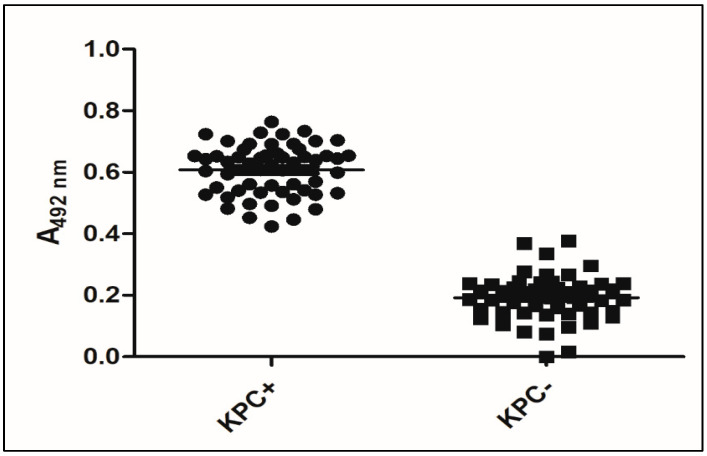
T-Student analysis of the KPC-ELISA-GMHB test verification. Caption: KPC+: KPC-like-producing samples. KPC-: isolates not producing KPC-like (*p*-value: <0.0001).

**Figure 7 microorganisms-11-01052-f007:**
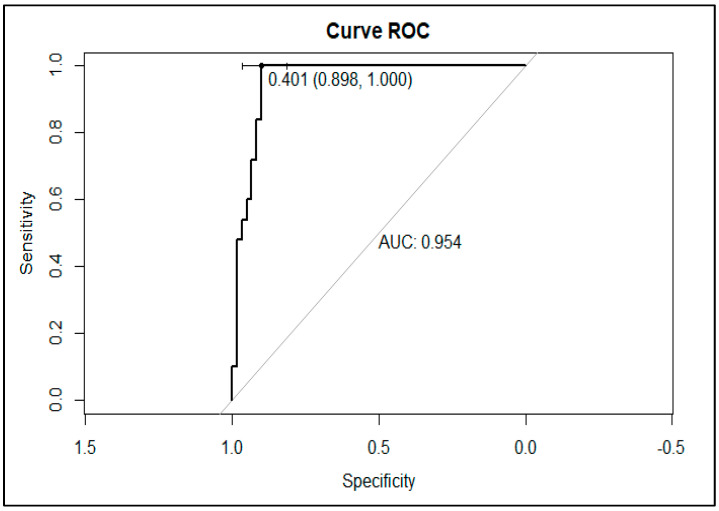
ROC curve of the performance of the ELISA-KPC-GMHB. This methodology obtained a sensitivity of 100% and a specificity of 89%, with 95% IC and a cut-off of 0.401.

**Table 1 microorganisms-11-01052-t001:** Isolates used for screening protocol.

	Bacterial Species	Type of Carbapenemases (Number of Isolates Tested)
Carbapenemase-producing isolates		
	*Klebsiella pneumoniae* (7)	KPC-2 (4)
		BKC-1 (3)
	*Pseudomonas aeruginosa* (3)	VIM-1 (1)
		GES-5 (1)
		SPM-1 (1)
	*Escherichia coli* (1)	NDM-1 (1)
Non-Carbapenemase-producing isolates		
	*Klebsiella pneumoniae* (1)	SHV-18 (1)
Non-β-lactamase producing isolates		
	*Escherichia coli* (1)	None
	Total:	13 isolates

**Table 2 microorganisms-11-01052-t002:** Isolates used for ELISA-KPC test verification.

Carbapenemase-Producing Isolates/Species		Type of Carbapenemase (Number of Isolates Tested)
KPC-producing isolates positive (N = 50)		
	*Klebsiella pneumoniae* (47)	
		KPC-2 (35)
		KPC-3 (7)
		KPC-7 (1)
		KPC-11 (1)
		KPC-33 (1)
		KPC-66 (1)
		KPC-144 (1)
	*Klebsiella oxytoca* (1)	
		KPC-2 (1)
	*Pseudomonas aeruginosa* (2)	
		KPC-2 (2)
Double-carbapenemase producer (N = 3)		
	*Klebsiella pneumoniae* (3)	
		KPC-2/BKC-1 (3)
Other carbapenemases (N = 24)		
	*Klebsiella pneumoniae* (12)	
		BKC-1 (6)
		BKC-2 (1)
		NDM-1 (1)
		OXA-48 (1)
		GES-5 (2)
		IMP-1 (1)
	*Pseudomonas aeruginosa* (8)	
		NDM-1 (1)
		IMP-1 (2)
		GIM-1 (2)
		SPM-1 (3)
	*Pseudomonas monteilli* (1)	
		VIM-1 (1)
	*Enterobacter* spp. (3)	
		NDM-1 (3)
Non-carbapenemase-producing isolates (N = 19)		
	*Klebsiella pneumoniae* (2)	
		OmpK-36 (1)
		OmpK (35/36) (1)
	*Pseudomonas aeruginosa* (16)	
		Non carbapenemase producer
	*Serratia marcescens* (1)	
		GES-16 (1)
ESBL-producing isolates (N = 3)		
	*Klebsiella pneumoniae* (3)	
		CTX-M8 (2)
		SHV-18 (1)
Isolates susceptible to carbapenems (N = 10)		
	*Klebsiella pneumoniae* (4)	
	*Escherichia coli* (2)	
	*Acinetobacter baumannii* (1)	
	*Proteus mirabilis* (1)	
	*Shigella flexneri* (1)	
	*Salmonella enterica* subsp. *Enterica* (1)	
		Total N = 109 isolates

**Table 3 microorganisms-11-01052-t003:** Results of the block titration. Average of three block titration assays to define optimal mouse and rabbit pAb concentrations to be used in the ELISA test. Legend: **A**: titration using PBS; **B**, titration using rKPC-2. * We used the 6.25 µg/mL mouse pAb dilution and also 6.25 µg/mL rabbit pAb.

**A**	**Rabbit (µg/mL)**			
**Mice (µg/mL)**	**25**	**12.5**	**6.25**	**3.125**
12.5	0.195	0.146	0.11	0.104
6.25	0.181	0.116	0.104 (*)	0.073
3.125	0.158	0.109	0.078	0.063
1.5625	0.154	0.095	0.07	0.057
0.78125	0.151	0.098	0.076	0.059
0.390625	0.146	0.091	0.066	0.057
0.1953125	0.147	0.1	0.073	0.058
0.09765625	0.149	0.096	0.078	0.06
**B**	**Rabbit (µg/mL)**			
**Mice (µg/mL)**	**25**	**12.5**	**6.25**	**3.125**
12.5	1.165	1.091	1.062	0.948
6.25	1.055	1.097	1.026 (*)	0.925
3.125	0.855	0.825	0.835	0.708
1.5625	0.833	0.714	0.666	0.571
0.78125	0.923	0.754	0.695	0.518
0.390625	0.936	0.695	0.722	0.57
0.1953125	0.87	0.829	0.682	0.609
0.09765625	0.988	0.871	0.737	0.576

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
