# Peer review of "Capture ELISA for KPC Detection in Gram-Negative Bacilli: Development and Standardisation"

_microorganisms, 2023, doi:10.3390/microorganisms11041052_

Round 1
Reviewer 1 Report
The manuscript entitled “Capture ELISA for KPC Detection in Gram-Negative Bacilli: Development and Standardisation” is well-described and is of interest. A standardization protocol for KPC detection using pAbs on an ELISA platform is described here for the first time. These results may pave the way to develop another similar and even improved method.
Below, I pinpointed minor suggestions for improvement:
-Lines 18-19: “Sample preparation methodologies showed variation in sensitivity (100%), specificity (89%).” This sentence is not clear. If the methodologies showed variation in sensitivity and specificity, I would expect to see a range of values regarding that variation and not only a single value, 100% or 89%. Please clarify or correct appropriately.
- Line 20: Delete the closing bracket at the end of the sentence.
-Lines 149-156: In protocols iii) and iv) it is not mentioned at the end if the pellet is suspended in PBS too. I guess it was, but that step is missing. You must insert that information.
- Figure 7: In the graphs or in the legend the respective protocols (DC, PD-G, PD-OD1, GMHB) must be indicated.
-Line 241: “The ELISA test detected all 53 KPC-producing isolates…” Here and then in Fig. 8 and 9, authors refer to ELISA test in general, however, it must be specified that they are referring to the GMHB ELISA test. Otherwise, it can not be clear to the readers which is the ELISA test now generally mentioned.
Lines 272 and 273: Avoid the repetition of “in addition” in these subsequent sentences.
Author Response
Manuscript ID: microorganisms-2306130
Capture ELISA for KPC Detection in Gram-Negative Bacilli: Development and Standardisation
Valencio, A, et al.
We appreciate all the suggestions that have improved our manuscript. Below is the point-by-point response to the reviewers` comments. All the modifications made in the manuscript are highlighted in yellow.
Answer to Reviewer 1:
-Lines 18-19: “Sample preparation methodologies showed variation in sensitivity (100%), specificity (89%).” This sentence is not clear. If the methodologies showed variation in sensitivity and specificity, I would expect to see a range of values regarding that variation and not only a single value, 100% or 89%. Please clarify or correct appropriately.
Answer: We rewrote the sentence to clarify this point. “The standardisation was performed using 109 previously characterized clinical isolates, showing 100% of sensitivity and 89% of specificity. The ELISA-KPC detected all isolates producing carbapenemases, including KPC variants displaying ESBL phenotype like KPC-33 and -66.”
- Line 20: Delete the closing bracket at the end of the sentence.
Answer: We deleted the bracket.
-Lines 149-156: In protocols iii) and iv) it is not mentioned at the end if the pellet is suspended in PBS too. I guess it was, but that step is missing. You must insert that information.
Answer: We included the following sentence at the end of both protocols to better clarify this point: “… the supernatant was discarded, and the pellet was resuspended in 500 µL 1% PBS.”
- Figure 7: In the graphs or in the legend the respective protocols (DC, PD-G, PD-OD1, GMHB) must be indicated.
Answer: We indicated the protocol in the legend.
-Line 241: “The ELISA test detected all 53 KPC-producing isolates…” Here and then in Fig. 8 and 9, authors refer to ELISA test in general, however, it must be specified that they are referring to the GMHB ELISA test. Otherwise, it can not be clear to the readers which is the ELISA test now generally mentioned.
Answer: We specified the ELISA-GMHB throughout the manuscript.
Lines 272 and 273: Avoid the repetition of “in addition” in these subsequent sentences.
Answer: We deleted one of the “in addition” to avoid repetition.
Reviewer 2 Report
the paper represents a well thought out and realistic procedure for the development and standardization of Capture ELISA for KPC Detection in Gram-Negative Bacilli. The methodology that was used is applicable in the development of kits for the ELISA test for individual pathogens that cannot be purchased in commercial packages and requires individual development. The procedure is clearly shown and the method of standardization is in full compliance with the GLP that we use when we have to do things like this.
Author Response
Manuscript ID: microorganisms-2306130
Capture ELISA for KPC Detection in Gram-Negative Bacilli: Development and Standardisation
Valencio, A, et al.
We appreciate all the suggestions that have improved our manuscript.
Reviewer 3 Report
This manuscript is well organized and is valuable for clinical diagnosis.
The language is fine with only several minor points.
Line 20 -66) should be -66?
Line 95 rKPC should be rKPC-2?
Line 143 the number should be in subscript in Na2HPO4...
All table in the manuscript should be in three-line table
Line 184 Recombinant KPC-2 protein should be the blakpc-2 amplicon
Figure 3 and 4 can be shown in one figure with A and B
Figure 6 is not a Figure, it should be a Table
Author Response
Manuscript ID: microorganisms-2306130
Capture ELISA for KPC Detection in Gram-Negative Bacilli: Development and Standardisation
Valencio, A, et al.
We appreciate all the suggestions that have improved our manuscript. Below is the point-by-point response to the reviewers` comments. All the modifications made in the manuscript are highlighted in yellow.
Answer to Reviewer 3:
Line 20 -66) should be -66?
Answer: We deleted the bracket.
Line 95 rKPC should be rKPC-2?
Answer: We changed rKPC to rKPC-2 throughout the manuscript.
Line 143 the number should be in subscript in Na2HPO4...
Answer: We subscripted the numbers.
All table in the manuscript should be in three-line table
Answer: We modified all the tables.
Line 184 Recombinant KPC-2 protein should be the blakpc-2 amplicon
Answer: We changed this throughout the manuscript.
Figure 3 and 4 can be shown in one figure with A and B
Answer: We put figures 3 and 4 together.
Figure 6 is not a Figure, it should be a Table
Answer: We changed “Figure 6” to “Table 6” in a three-line table.